# Bridging the Gap: From Post Hoc Explanations to Inherently Interpretable Models for Medical Imaging

**Shantanu Ghosh** [1]   **Ke Yu** [2]   **Forough Arabshahi** [3]   **Kayhan Batmanghelich** [1]

## Abstract

ML model design either starts with an interpretable model or a Blackbox (BB) and explains it post hoc. BB models are flexible but difficult to explain, while interpretable models are inherently explainable. Yet, interpretable models require extensive ML knowledge and tend to be less flexible and underperforming than their BB variants. This paper aims to blur the distinction between a post hoc explanation of a BB and constructing interpretable models. Beginning with a BB, we iteratively *carve out* a mixture of interpretable experts and a *residual network*. Each interpretable model specializes in a subset of samples and explains them using First Order Logic (FOL). We route the remaining samples through a flexible residual. We repeat the method on the residual network until all the interpretable models explain the desired proportion of data. Our extensive experiments show that our approach (1) identifies a diverse set of instance-specific concepts without compromising the performance of the BB, (2) identifies the relatively "harder" samples to explain via residuals, and (3) is transferred to an unknown target domain with limited data efficiently. The code is uploaded at: https://github.com/batmanlab/MICCAI-2023-Route-interpret-repeat-CXRs.

## 1. Introduction

Model explainability is essential in high-stakes applications of AI, *e.g.,* healthcare. While BB models (*e.g.,* Deep Learning) offer flexibility and modular design, post hoc explanation is prone to confirmation bias (Wan et al., 2022), lack of fidelity to the original model (Adebayo et al., 2018), and insufficient mechanistic explanation of the decision-making process (Rudin, 2019). Interpretable-by-design models overcome those issues but tend to be less flexible than BB models and demand substantial expertise to design. Using a post hoc explanation or adopting an inherently interpretable model is a mutually exclusive decision to be made at the initial phase of AI model design. This paper blurs the line on that dichotomous model design.

The literature on post hoc explanations is extensive. This includes model attributions ( (Simonyan et al., 2013; Selvaraju et al., 2017)), counterfactual approaches (Abid et al., 2021; Singla et al., 2019), and distillation methods (Alharbi et al., 2021; Cheng et al., 2020). Those methods either identify key input features that contribute the most to the network's output (Shrikumar et al., 2016), generate input perturbation to flip the network's output (Samek et al., 2016; Montavon et al., 2018), or estimate simpler functions to approximate the network output locally. Post hoc methods preserve the flexibility and performance of the BB but suffer from a lack of fidelity and mechanistic explanation of the network output (Rudin, 2019). Without a mechanistic explanation, the recourse to a model's undesirable behavior is unclear. Interpretable models are alternative designs to the BB without many such drawbacks. For example, modern interpretable methods highlight human understandable *concepts* that contribute to the downstream prediction.

Several families of interpretable models exist for a long time, such as the rule-based approach and generalized additive models (Hastie & Tibshirani, 1987; Letham et al., 2015; Breiman et al., 1984). They primarily focus on tabular data. Such models for high-dimensional data (*e.g.,* images) primarily rely on projecting to a lower dimensional human understandable *concept* or *symbolic* space (Koh et al., 2020) and predicting the output with an interpretable classifier. Despite their utility, the current State-Of-The-Art (SOTA) are limited in design; for example, they do not model the interaction between the concepts except for a few exceptions (Ciravegna et al., 2021; Barbiero et al., 2022), offering limited reasoning capabilities and robustness. Furthermore, if a portion of the samples does not fit the template design

---

[1]Department of Electrical and Computer Engineering, Boston University, MA, USA [2]Intelligent Systems Program, University of Pittsburgh, PA, USA [3]MetaAI, MenloPark, CA, USA. Correspondence to: Shantanu Ghosh <shawn24@bu.edu>.

*Workshop on Interpretable ML in Healthcare at International Conference on Machine Learning (ICML)*, Honolulu, Hawaii, USA.

of the interpretable model, they do not offer any flexibility, compromising performance.

**Our contributions:** We propose an interpretable method to achieve the best of both worlds: not sacrificing BB performance similar to post hoc explainability while still providing actionable interpretation. We hypothesize that a BB encodes several interpretable models, each applicable to a different portion of data. Thus, a single interpretable model may be insufficient to explain all samples. We construct a hybrid neuro-symbolic model by progressively *carving out* a mixture of interpretable models and a *residual network* from the given BB. We coin the term *expert* for each interpretable model, as they specialize over a subset of data. All the interpretable models are termed a "Mixture of Interpretable Experts" (MoIE). Our design identifies a subset of samples and *routes* them through the interpretable models to explain the samples with FOL, providing basic reasoning on concepts from the BB. The remaining samples are routed through a flexible residual network. On the residual network, we repeat the method until MoIE explains the desired proportion of data. Also, we apply our method to a real-life CXR dataset. Due to class imbalance in large CXR datasets, early interpretable models tend to cover all samples with disease present while ignoring disease subgroups and pathological heterogeneity. In this context, we propose MoIE for CXRs (MoIE-CXR) to address this problem by estimating the class-stratified coverage from the total data coverage. We assume that MoIE-CXR discovers the sample-specific domain-invariant concepts for diverse disease subtypes and pathological patterns. This process is analogous to how radiologists search for patterns of anatomical changes to detect abnormalities in medical images, and subsequently apply logical rules to arrive at specific diagnoses. So, we apply MoIE-CXR to a target domain with limited training data in a transfer learning setup. The target domain lacks concept-level annotation since they are expensive. Hence, we learn a concept detector in the target domain with a pseudo labeling approach (Lee et al., 2013) and finetune MoIE-CXR. Our work is the first to apply concept-based methods to CXRs and transfer them between domains.

## 2. Method

**Notation:** Assume we have a dataset $\{\mathcal{X}, \mathcal{Y}, \mathcal{C}\}$, where $\mathcal{X}$, $\mathcal{Y}$, and $\mathcal{C}$ are the input images, class labels, and human interpretable attributes, respectively. $f^0 : \mathcal{X} \to \mathcal{Y}$, is our pre-trained initial BB model. We assume that $f^0$ is a composition $h^0 \circ \Phi$, where $\Phi : \mathcal{X} \to \mathbb{R}^l$ is the image embeddings and $h^0 : \mathbb{R}^l \to \mathcal{Y}$ is a transformation from the embeddings, $\Phi$, to the class labels. We denote the learnable function $t : \mathbb{R}^l \to \mathcal{C}$, projecting the image embeddings to the concept space (Ghosh et al., 2023b;a;c). The concept space is the space spanned by the attributes $\mathcal{C}$. Thus, function $t$ outputs

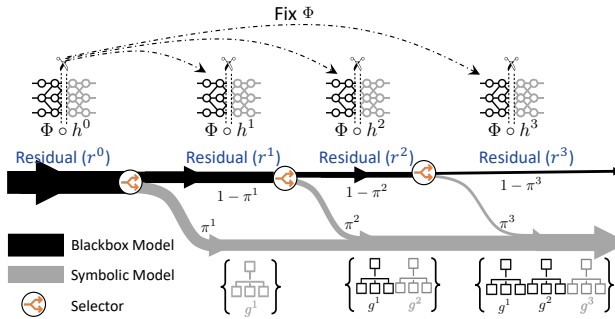

*Figure 1.* Schematic view of our method. Note that $f^k(.) = h^k(\Phi(.))$. At iteration $k$, the selector *routes* each sample either towards the expert $g^k$ with probability $\pi^k(.)$ or the residual $r^k = f^{k-1} - g^k$ with probability $1 - \pi^k(.)$. $g^k$ generates FOL-based explanations for the samples it covers. Note $\Phi$ is fixed across iterations.

*Table 1.* Datasets and BBs'.

| DATASET | BB | # EXPERTS |
|---|---|---|
| HAM1000 (Tschandl et al., 2018) | INCEPTION (Szegedy et al., 2015) | 6 |
| SIIM-ISIC (Rotemberg et al., 2021) | INCEPTION (Szegedy et al., 2015) | 6 |
| CARDIOMEGALY IN MIMIC-CXR (Johnson et al.) | DENSENET121 (Huang et al., 2017) | 4 |
| EDEMA IN MIMIC-CXR (Johnson et al.) | DENSENET121 (Huang et al., 2017) | 5 |
| EFFUSION IN MIMIC-CXR (Johnson et al.) | DENSENET121 (Huang et al., 2017) | 5 |
| PNEUMONIA IN MIMIC-CXR (Johnson et al.) | DENSENET121 (Huang et al., 2017) | 4 |
| PNEUMOTHORAX IN MIMIC-CXR (Johnson et al.) | DENSENET121 (Huang et al., 2017) | 5 |

a scalar value representing a concept for each input image.

**Method Overview:** Figure 1 summarizes our approach. We iteratively carve out an interpretable model from the given BB. Each iteration yields an interpretable model (the downward grey paths in Figure 1) and a residual (the straightforward black paths in Figure 1). We start with the initial BB $f^0$. At iteration $k$, we distill the BB from the previous iteration $f^{k-1}$ into a neuro-symbolic interpretable model, $g^k : \mathcal{C} \to \mathcal{Y}$. Our $g$ is flexible enough to be any interpretable models (Yuksekgonul et al., 2022; Koh et al., 2020; Barbiero et al., 2022). The *residual* $r^k = f^{k-1} - g^k$ emphasizes the portion of $f^{k-1}$ that $g^k$ cannot explain. We then approximate $r^k$ with $f^k = h^k \circ \Phi$. $f^k$ will be the BB for the subsequent iteration and be explained by the respective interpretable model. A learnable gating mechanism, denoted by $\pi^k : \mathcal{C} \to \{0, 1\}$ (shown as the *selector* in Figure 1) routes an input sample towards either $g^k$ or $r^k$. The thickness of the lines in Figure 1 represents the samples covered by the interpretable models (grey line) and the residuals (black line). With every iteration, the cumulative coverage of the interpretable models increases, but the residual decreases. We name our method *route, interpret* and *repeat*.

**Algorithm 1** Finetuning to an unseen domain.

1: **Input:** Learned selectors, experts, and final residual from source domain: $\{\pi_s^k, g_s^k\}_{k=1}^K$ and $f_s^K$ respectively, with $K$ as the number of experts to transfer. BB of the source domain: $f_s^0 = h_s^0(\Phi_s)$. Source data: $\mathcal{D}_s = \{\mathcal{X}_s, \mathcal{C}_s, \mathcal{Y}_s\}$. Target data: $\mathcal{D}_t = \{\mathcal{X}_t, \mathcal{Y}_t\}$. Target coverages $\{\tau_k\}_{k=1}^K$.
2: **Output:** Experts $\{\pi_t^k, g_t^k\}_{k=1}^K$ and final residual $f_t^K$ of the target domain.
3: Randomly select $n_t \ll N_t$ samples out of $N_t = |\mathcal{D}_t|$.
4: Compute the pseudo concepts for the correctly classified samples in the target domain using $f_s^0$, as, $\boldsymbol{c_t^i} = t_s(\Phi_s(\boldsymbol{x_s}^i))$ *s.t.*, $y_t^i = f_s^0(\boldsymbol{x_t}^i)$, $i = 1 \cdots n_t$
5: Learn the projection function $t_t$ for target domain semi-supervisedly (Lee et al., 2013) using the pseudo labeled samples $\{\boldsymbol{x}_t^i, \boldsymbol{c}_t^i\}_{i=1}^{n_t}$ and unlabeled samples $\{\boldsymbol{x}_t^i\}_{i=1}^{N_t - n_t}$.
6: Complete the triplet for the target domain $\{\mathcal{X}_t, \mathcal{C}_t, \mathcal{Y}_t\}$, where $\boldsymbol{c}_t^i = t_t(\Phi_s(\boldsymbol{x}_t^i))$, $i = 1 \cdots N_t$.
7: Finetune $\{\pi_s^k, g_s^k\}_{k=1}^K$ and $f_s^K$ to obtain $\{\pi_t^k, g_t^k\}_{k=1}^K$ and $f_t^K$ using equations 5, 3 and 4 respectively for 5 epochs. $\{\pi_t^k, g_t^k\}_{k=1}^K$ and $\{\{\pi_t^k, g_t^k\}_{k=1}^K, f_t^K\}$ represents MoIE-CXR and MoIE-CXR + R for the target domain.

## 2.1. Neuro-Symbolic Knowledge Distillation

Knowledge distillation in our method involves 3 parts: (1) a series of trainable selectors, *routing* each sample through the interpretable models and the residual networks, (2) a sequence of learnable neuro-symbolic interpretable models, each providing FOL explanations to *interpret* the BB, and (3) *repeating* with Residuals for the samples that cannot be explained with their interpretable counterparts. We detail each component below.

### 2.1.1. THE SELECTOR FUNCTION

As the first step of our method, the selector $\pi^k$ *routes* the $j^{th}$ sample through the interpretable model $g^k$ or residual $r^k$ with probability $\pi^k(\boldsymbol{c_j})$ and $1 - \pi^k(\boldsymbol{c_j})$ respectively, where $k \in [0, K]$, with $K$ being the number of iterations. We define the empirical coverage of the $k^{th}$ iteration as $\zeta(\pi^k) = \frac{1}{m} \sum_{j=1}^m \pi^k(\boldsymbol{c_j})$, the empirical mean of the samples selected by the selector for the associated interpretable model $g^k$, with $m$ being the total number of samples in the training set. Thus, the entire selective risk is:

$$\mathcal{R}^k(\pi^k, g^k) = \frac{\frac{1}{m} \sum_{j=1}^m \mathcal{L}_{(g^k, \pi^k)}^k (\boldsymbol{x_j}, \boldsymbol{c_j})}{\zeta(\pi^k)}, \quad (1)$$

where $\mathcal{L}_{(g^k, \pi^k)}^k$ is the optimization loss used to learn $g^k$ and $\pi^k$ together, discussed in Section 2.1.2. For a given coverage

of $\tau^k \in (0, 1]$, we solve the following optimization problem:

$$\theta_{s^k}^*, \theta_{g^k}^* = \underset{\theta_{s^k}, \theta_{g^k}}{\arg \min} \mathcal{R}^k \Big( \pi^k(.; \theta_{s^k}), g^k(.; \theta_{g^k}) \Big)$$
$$\text{s.t.} \quad \zeta\big(\pi^k(.; \theta_{s^k})\big) \geq \tau^k, \quad (2)$$

where $\theta_{s^k}^*, \theta_{g^k}^*$ are the optimal parameters at iteration $k$ for the selector $\pi^k$ and the interpretable model $g^k$ respectively. In this work, $\pi$s' of different iterations are neural networks with sigmoid activation. At inference time, the selector routes the $j^{th}$ sample with concept vector $\boldsymbol{c_j}$ to $g^k$ if and only if $\pi^k(\boldsymbol{c_j}) \geq 0.5$ for $k \in [0, K]$.

### 2.1.2. NEURO-SYMBOLIC INTERPRETABLE MODELS

In this stage, we design interpretable model $g^k$ of $k^{th}$ iteration to *interpret* the BB $f^{k-1}$ from the previous $(k-1)^{th}$ iteration by optimizing the following loss function:

$$\mathcal{L}_{(g^k, \pi^k)}^k(\boldsymbol{x_j}, \boldsymbol{c_j}) = \underbrace{\ell\Big(f^{k-1}(\boldsymbol{x_j}), g^k(\boldsymbol{c_j})\Big)\pi^k(c_j)}_{\substack{\text{trainable component} \\ \text{for current iteration } k}} \underbrace{\prod_{i=1}^{k-1} \big(1 - \pi^i(\boldsymbol{c_j})\big)}_{\substack{\text{fixed component trained} \\ \text{in the previous iterations}}},$$
$$(3)$$

where the term $\pi^k(\boldsymbol{c_j}) \prod_{i=1}^{k-1} \big(1 - \pi^i(\boldsymbol{c_j})\big)$ denotes the probability of $j^{th}$ sample being routed through the interpretable model $g^k$. It is the probability of the sample going through the residuals for all the previous iterations from 1 through $k-1$ (*i.e.*, $\prod_{i=1}^{k-1} \big(1 - \pi^i(\boldsymbol{c_j})\big)$) times the probability of going through the interpretable model at iteration $k$ $\big($*i.e.*, $\pi^k(\boldsymbol{c_j})\big)$. Refer to Figure 1 for an illustration. We learn $\pi^1, \ldots \pi^{k-1}$ in the prior iterations and are not trainable at iteration $k$. As each interpretable model $g^k$ specializes in explaining a specific subset of samples (denoted by coverage $\tau$), we refer to it as an *expert*. We use SelectiveNet's (Geifman & El-Yaniv, 2019) optimization method to optimize Equation (5) since selectors need a rejection mechanism to route samples through residuals. Appendix A.4 details the optimization procedure in Equation (3). We refer to the interpretable experts of all the iterations as a "Mixture of Interpretable Experts" (MoIE) cumulatively after training. Furthermore, we utilize E-LEN, *i.e.,* a Logic Explainable Network (Ciravegna et al., 2023) implemented with an Entropy Layer as first layer (Barbiero et al., 2022) as the interpretable symbolic model $g$ to construct First Order Logic (FOL) explanations of a given prediction.

### 2.1.3. THE RESIDUALS

The last step is to *repeat* with the residual $r^k$, as $r^k(\boldsymbol{x_j}, \boldsymbol{c_j}) = f^{k-1}(\boldsymbol{x_j}) - g^k(\boldsymbol{c_j})$. We train $f^k = h^k\big(\Phi(.)\big)$ to approximate the residual $r^k$, creating a new BB $f^k$ for the next iteration $(k+1)$. This step is necessary to specialize $f^k$ over samples not covered by $g^k$. Optimizing the following

loss function yields $f^k$ for the $k^{th}$ iteration:

$$\mathcal{L}_f^k(\boldsymbol{x_j}, \boldsymbol{c_j}) = \underbrace{\ell\big(r^k(\boldsymbol{x_j}, \boldsymbol{c_j}), f^k(\boldsymbol{x_j})\big)}_{\substack{\text{trainable component} \\ \text{for iteration } k}} \underbrace{\prod_{i=1}^{k}\big(1 - \pi^i(\boldsymbol{c_j})\big)}_{\substack{\text{non-trainable component} \\ \text{for iteration } k}}$$

(4)

Notice that we fix the embedding $\Phi(.)$ for all the iterations. Due to computational overhead, we only finetune the last few layers of the BB ($h^k$) to train $f^k$. At the final iteration $K$, our method produces a MoIE and a Residual, explaining the interpretable and uninterpretable components of the initial BB $f^0$, respectively. Appendix A.5 describes the training procedure of our model, the extraction of FOL, and the architecture of our model at inference.

**Selecting number of iterations $K$:** We follow two principles to select the number of iterations $K$ as a stopping criterion: 1) Each expert should have enough data to be trained reliably ( coverage $\zeta^k$). If an expert covers insufficient samples, we stop the process. 2) If the final residual ($r^K$) underperforms a threshold, it is not reliable to distill from the BB. We stop the procedure to ensure that overall accuracy is maintained.

## 2.2. Creating MoIE-CXR from MoIE

Most of the real-life CXR datasets suffer from class imbalance where the samples without the disease outnumber the samples with the disease. Due to class imbalance early interpretable models in MoIE tend to cover all samples with disease present while ignoring disease subgroups and pathological heterogeneity. This section handles this problem by introducing stratified coverage to learn the selectors in MoIE and create the Mixture of Interpretable Experts for chest-X-Rays (MoIE-CXR). We assume $m$, as the number of class labels. This paper focuses on binary classification (having or not having a disease), so $m = 2$ and $\mathcal{Y} \in \{0, 1\}$. Yet, it can be extended to multiclass problems easily.

### 2.2.1. HANDLING CLASS IMBALANCE IN CXRs

For an iteration $k$, we first split the given coverage $\tau^k$ to stratified coverages per class as $\{\tau_m^k = w_m \cdot \tau^k; w_m = N_m/N, \forall m\}$, where $w_m$ denotes the fraction of samples belonging to the $m^{th}$ class; $N_m$ and $N$ are the samples of $m^{th}$ class and total samples, respectively.

### 2.2.2. LEARNING SELECTORS FOR CXRs

As per in Section 2.1.1, at iteration $k$, the selector $\pi^k$ *routes* $i^{th}$ sample to the expert ($g^k$) or residual ($r^k$) with probability $\pi^k(\boldsymbol{c_i})$ and $1 - \pi^k(\boldsymbol{c_i})$ respectively. For stratified coverages $\{\tau_m^k, \forall m\}$, we learn $g^k$ and $\pi^k$ jointly by chang-

ing the Equation (5) to the following optimization problem:

$$\theta_{s^k}^*, \theta_{g^k}^* = \underset{\theta_{s^k}, \theta_{g^k}}{\arg\min} \, \mathcal{R}^k\Big(\pi^k(.; \theta_{s^k}), g^k(.; \theta_{g^k})\Big)$$

$$\text{s.t.} \quad \zeta_m\big(\pi^k(.; \theta_{s^k})\big) \geq \tau_m^k \;\; \forall m, \qquad (5)$$

where $\theta_{s^k}^*, \theta_{g^k}^*$ are the optimal parameters for $\pi^k$ and $g^k$, respectively. We modify the overall selective risk $\mathcal{R}^k$ from Equation (1) to $\mathcal{R}^k(\pi^k, g^k) = \sum_m \frac{\frac{1}{N_m} \sum_{i=1}^{N_m} \mathcal{L}_{(g^k, \pi^k)}^k(\boldsymbol{x_i}, \boldsymbol{c_i})}{\zeta_m(\pi^k)}$, where $\zeta_m(\pi^k) = \frac{1}{N_m} \sum_{i=1}^{N_m} \pi^k(\boldsymbol{c_i})$ is the empirical mean of samples of $m^{th}$ class selected by the selector for the associated expert $g^k$.

### 2.2.3. LEARNING EXPERTS AND RESIDUALS IN CXRs

Finally, we learn the experts and residuals in MoIE-CXR using Equation (3) and Equation (4) respectively.

## 2.3. Finetuning MoIE-CXR efficiently to an unseen domain with limited training samples

We assume the MoIE-CXR-identified concepts to be generalizable to an unseen domain. So, we learn the projection $t_t$ for the target domain and compute the pseudo concepts using SSL (Lee et al., 2013). Next, we transfer the selectors, experts, and final residual ($\{\pi_s^k, g_s^k\}_{k=1}^K$ and $f_s^K$) from the source to a target domain with limited labeled data and computational cost. Algorithm 1 details the procedure.

## 3. Related Work

**Post hoc explanations:** Post hoc explanations retain the flexibility and performance of the BB. It includes feature attribution (Simonyan et al., 2013; Smilkov et al., 2017; Binder et al., 2016) and counterfactual approaches (Singla et al., 2019; Abid et al., 2021). For example, feature attribution methods associate a measure of importance to features (e.g., pixels) that is proportional to the feature's contribution to BB's predicted output. Many methods were proposed to estimate the importance measure, including gradient-based methods (Selvaraju et al., 2017; Sundararajan et al., 2017), game-theoretic approach (Lundberg & Lee, 2017). The post hoc approaches suffer from a lack of fidelity to input (Adebayo et al., 2018) and ambiguity in explanation due to a lack of correspondence to human-understandable concepts. Recently, Posthoc Concept Bottleneck models (PCBMs) (Yuksekgonul et al., 2022) learn the concepts from a trained BB embedding and use an interpretable classifier for classification. Also, they fit a residual in their hybrid variant (PCBM-h) to mimic the performance of the BB. We will compare against the performance of the PCBMs method. Another major shortcoming is that, due to a lack of mechanistic explanation, post hoc explanations do not provide

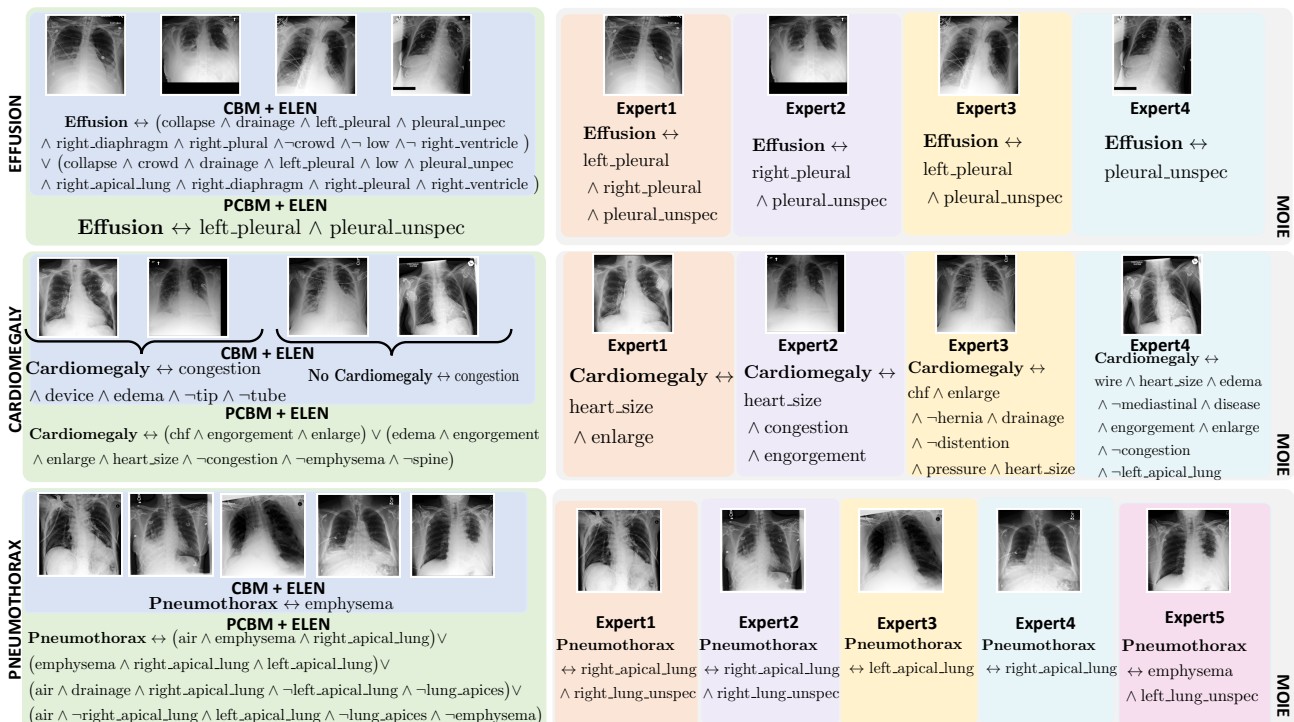

*Figure 2.* Qualitative comparison of MoIE discovered concepts with the baselines.

a recourse when an undesirable property of a BB is identified. Interpretable-by-design provides a remedy to those issues (Rudin, 2019).

**Concept-based interpretable models:** Our approach follows the concept-based interpretable models. Such methods provide a mechanistically interpretable prediction that is a function of human-understandable concepts. The concepts are usually extracted from the activation of the middle layers of the Neural Network (bottleneck). Examples include Concept Bottleneck models (CBMs) (Koh et al., 2020), antehoc concept decoder (Sarkar et al., 2021), and a high-dimensional Concept Embedding model (CEMs) (Zarlenga et al., 2022) that uses high dimensional concept embeddings to allow extra supervised learning capacity and achieves SOTA performance in the interpretable-by-design class. Most concept-based interpretable models do not model the interaction between concepts and cannot be used for reasoning. An exception is E-LEN (Barbiero et al., 2022) which uses an entropy-based approach to derive explanations in terms of FOL using the concepts. The underlying assumption of those methods is that one interpretable function can explain the entire set of data, which can limit flexibility and consequently hurt the performance of the models. Our approach relaxes that assumption by allowing multiple interpretable functions and a residual. Each function is appropriate for a portion of the data, and a small portion of the data

is allowed to be uninterpretable by the model (*i.e.,* residual). We will compare our method with CBMs, CEMs, and their E-LEN-enhanced variants.

## 4. Experiments

We perform experiments on the skin and CXR datasets to show that 1) our method captures a diverse set of concepts, 2) the performance of the residuals degrades over successive iterations as they cover "harder" instances, 3) our method does not compromise the performance of the BB, 4) our method is finetuned well to an unseen domain with minimal computation.

**Experimental Details.** We apply MoIE to HAM10000 dataset. We then use MoIE on SIIM-ISIC as a real-world transfer learning setting (Yuksekgonul et al., 2022), with the BB trained on HAM10000 and evaluated on a subset of the SIIM-ISIC Melanoma Classification. We deploy MoIE-CXR to classify cardiomegaly, effusion, edema, pneumonia, and pneumothorax in the MIMIC-CXR dataset, considering each to be a separate binary classification problem. We repeat our method until MoIE covers at least 90% of samples or the final residual's accuracy falls below 70%. Furthermore, we only include concepts as input to $g$ if their validation accuracy or auroc exceeds a certain threshold (in all of our experiments, we fix 0.7 or 70% as the threshold of vali-

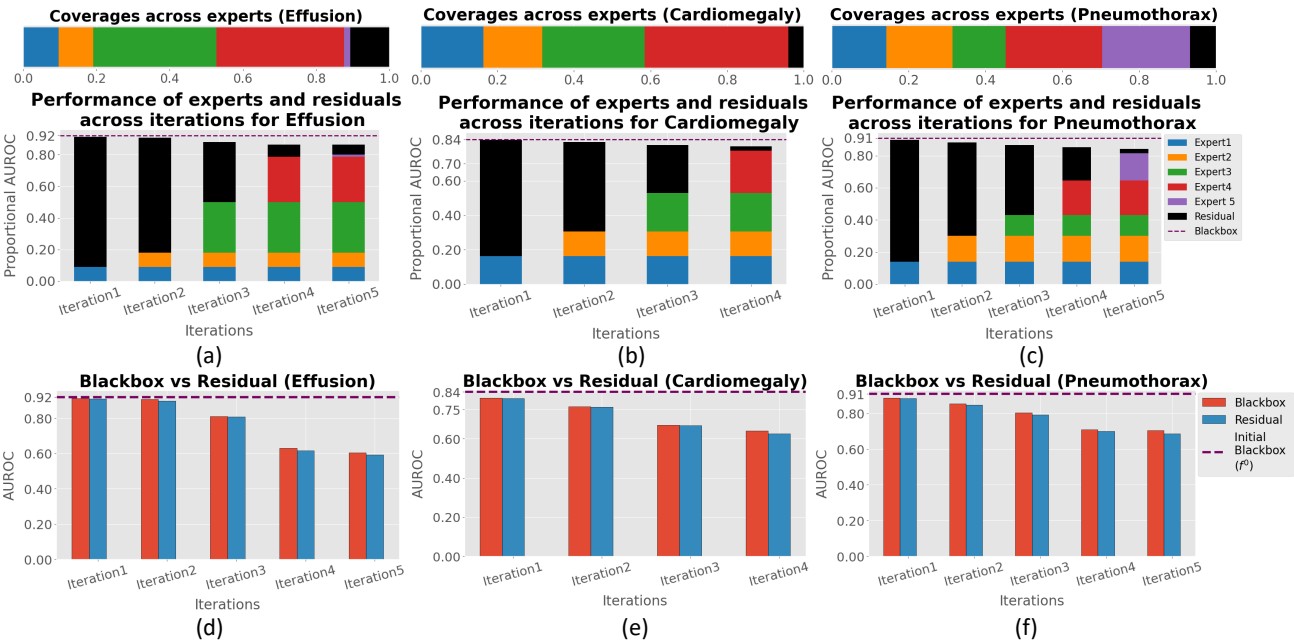

Figure 3. Performance of experts and residuals across iterations. **(a-c):** Coverage and proportional AUROC of the experts and residuals. **(d-f):** Routing the samples covered by MoIE to the initial $f^0$, we compare the performance of the residuals with $f^0$.

dation auroc or accuracy). Refer to Table 1 for the datasets and BBes experimented with. For all the BBes, we flatten the feature maps from the last convolutional block to extract the concepts. For VITs, we use the image embeddings from the transformer encoder to perform the same. We use SIIM-ISIC as a real-world transfer learning setting, with the BB trained on HAM10000 and evaluated on a subset of the SIIM-ISIC Melanoma Classification dataset (Yuksekgonul et al., 2022). Appendix A.6 and Appendix A.7 expand on the datasets and hyperparameters respectively.

**Baselines:** We compare our methods to two concept-based baselines – 1) interpretable-by-design and 2) posthoc. They consist of two parts: a) a concept predictor $\Phi : \mathcal{X} \to \mathcal{C}$, predicting concepts from images; and b) a label predictor $g : \mathcal{C} \to \mathcal{Y}$, predicting labels from the concepts. The end-to-end CEMs and sequential CBMs serve as interpretable-by-design baselines. Similarly, PCBM and PCBM-h serve as post hoc baselines. The $\Phi$ includes all layers till the last convolution block. The standard CBM and PCBM models do not show how the concepts are composed to make the label prediction. So, we create CBM + E-LEN, PCBM + E-LEN, and PCBM-h + E-LEN by using the same $g$ of MOIE replacing the standard classifiers of CBM and PCBM. We train the $\Phi$ and $g$ in these new baselines to sequentially generate FOLs (Barbiero et al., 2022). Due to the unavailability of concept annotations, we extract the concepts from the Derm7pt dataset (Kawahara et al., 2018) using the pretrained embeddings of the BB (Yuksekgonul et al., 2022) for

HAM10000. Thus, we do not have interpretable-by-design baselines for HAM10000 and ISIC.

## 4.1. Results

### 4.1.1. EXPERT DRIVEN EXPLANATIONS

Figure 2 illustrates the FOL explanations for MIMIC-CXR. Recall that the experts ($g$) in MoIE-CXR and the baselines are ELEN (Barbiero et al., 2022), attributing attention weights to each concept. A concept with high attention weight indicates its high predictive significance. With a single $g$, the baselines rank the concepts in accordance with the identical order of attention weights for all the samples in a class, yielding a generic FOL for that class. In Fig. 2, the baseline PCBM + ELEN uses *left_pleural* and *pleural_unspec* to identify effusion for all four samples. MoIE-CXR deploys multiple experts, learning to specialize in distinct subsets of a class. So different interpretable models in MoIE assign different attention weights to capture instance-specific concepts unique to each subset. In Fig. 2 expert 2 relies on *right_pleural* and *pleural_unspec*, but expert 4 relies only on *pleural_unspec* to classify effusion. Figure 6 of Appendix A.9 illustrates FOLs for pneumonia and edema of MIMIC-CXR. Figure 7 in Appendix A.9 shows such diverse local instance-specific explanations for MoIE for HAM10000 (*top*) and ISIC (*bottom*).

The results show that the learned experts can provide more

*Table 2.* MoIE-CXR does not compromise the performance of BB. We provide the mean and standard errors of AUROC over five random seeds. For MoIE-CXR, we also report the percentage of test set samples covered by all experts as "*Coverage*". We boldfaced our results and BB.

| **Model** | Effusion | Cardiomegaly | Edema | Pneumonia | Pneumothorax |
|---|---|---|---|---|---|
| BB (BB) | **0.92** | **0.84** | **0.89** | **0.79** | **0.91** |
| **INTERPRETABLE BY DESIGN** | | | | | |
| CEM (Zarlenga et al., 2022) | $0.83_{\pm 1e-4}$ | $0.75_{\pm 1e-4}$ | $0.77_{\pm 2e-4}$ | $0.62_{\pm 4e-4}$ | $0.76_{\pm 3e-4}$ |
| CBM (Sequential) (Koh et al., 2020) | $0.78_{\pm 1e-4}$ | $0.72_{\pm 1e-4}$ | $0.77_{\pm 5e-4}$ | $0.60_{\pm 1e-3}$ | $0.75_{\pm 6e-4}$ |
| CBM + ELEN (Koh et al., 2020; Barbiero et al., 2022) | $0.81_{\pm 1e-4}$ | $0.72_{\pm 1e-4}$ | $0.79_{\pm 5e-4}$ | $0.62_{\pm 8e-4}$ | $0.75_{\pm 6e-4}$ |
| **POSTHOC** | | | | | |
| PCBM (Yuksekgonul et al., 2022) | $0.88_{\pm 1e-4}$ | $0.81_{\pm 1e-4}$ | $0.82_{\pm 1e-4}$ | $0.72_{\pm 1e-4}$ | $0.85_{\pm 7e-4}$ |
| PCBM-h (Yuksekgonul et al., 2022) | $0.90_{\pm 1e-4}$ | $0.83_{\pm 1e-4}$ | $0.85_{\pm 1e-4}$ | $0.77_{\pm 1e-4}$ | $0.89_{\pm 7e-4}$ |
| PCBM + ELEN (Yuksekgonul et al., 2022; Barbiero et al., 2022) | $0.90_{\pm 1e-4}$ | $0.82_{\pm 1e-4}$ | $0.85_{\pm 1e-4}$ | $0.75_{\pm 1e-4}$ | $0.85_{\pm 6e-4}$ |
| PCBM-h + ELEN (Yuksekgonul et al., 2022; Barbiero et al., 2022) | $0.91_{\pm 1e-4}$ | $0.83_{\pm 1e-4}$ | $0.87_{\pm 1e-4}$ | $0.77_{\pm 1e-4}$ | $0.90_{\pm 1e-4}$ |
| **OURS** | | | | | |
| MoIE-CXR $^{(Coverage)}$ | $\mathbf{0.93}^{(0.90)}_{\pm 1e-4}$ | $\mathbf{0.85}^{(0.96)}_{\pm 1e-4}$ | $\mathbf{0.91}^{(0.92)}_{\pm 1e-4}$ | $\mathbf{0.80}^{(0.97)}_{\pm 1e-4}$ | $\mathbf{0.91}^{(0.93)}_{\pm 2e-4}$ |
| MoIE-CXR+R | $\mathbf{0.91}_{\pm 1e-4}$ | $\mathbf{0.82}_{\pm 1e-4}$ | $\mathbf{0.88}_{\pm 1e-4}$ | $\mathbf{0.78}_{\pm 1e-4}$ | $\mathbf{0.90}_{\pm 2e-4}$ |

*Table 3.* MoIE does not hurt the performance of the original BB for HAM10000. We boldface our results. We also mention the "Coverage" for MoIE.

| MODEL | DATASET | |
|---|---|---|
| | HAM10000 | SIIM-ISIC |
| BB | 0.96 | 0.85 |
| **POSTHOC** | | |
| PCBM (Yuksekgonul et al., 2022) | 0.93 ± 0.00 | 0.71 ± 0.01 |
| PCBM-h (Yuksekgonul et al., 2022) | 0.95 ± 0.00 | 0.79 ± 0.05 |
| PCBM + E-LEN (Yuksekgonul et al., 2022; Barbiero et al., 2022) | 0.73 ± 0.01 | |
| PCBM-h + E-LEN (Yuksekgonul et al., 2022; Barbiero et al., 2022) | 0.94 ± 0.02 0.95 ± 0.03 | 0.82 ± 0.05 |
| **OURS** | | |
| MoIE (COVERAGE) | **0.95 ± 0.00 (0.9)** | **0.84 ± 0.00 (0.94)** |
| MoIE + RESIDUAL | **0.92 ± 0.00** | **0.82 ± 0.01** |

precise explanations at the subject level using the concepts, increasing confidence and trust in clinical use.

### 4.1.2. IDENTIFICATION OF HARDER SAMPLES BY SUCCESSIVE RESIDUALS

Fig. 3 (a-c) reports the proportional AUROC of the experts and the residuals per iteration. The proportional AUROC is the AUROC of that model times the empirical coverage, $\zeta^k$, the mean of the samples routed to the model by the respective selector ($\pi^k$). According to Fig. 3a in iteration 1, the residual (black bar) contributes more to the proportional AUROC than the expert1 (blue bar) for effusion with both achieving a cumulative proportional AUROC $\sim 0.92$. All the final experts collectively extract the entire interpretable component from BB $f^0$ in the final iteration, resulting in their more significant contribution to the cumulative performance. In subsequent iterations, the proportional AUROC decreases as the experts are distilled from the BB of the

previous iteration. The BB is derived from the residual that performs progressively worse with each iteration. The residual of the final iteration covers the "hardest" samples. Tracing these samples back to the original BB $f^0$, $f^0$ underperforms on these samples (Fig. 3 (d-f)) as the residual. Figure 8 and Figure 9 of Appendix A.10 demonstrates similar phenomena for other diseases (pneumonia and edema) of MIMIC-CXR and HAM10000 respectively.

### 4.1.3. QUANTITATIVE COMPARISON WITH THE BLACKBOX AND BASELINE

Table 2 shows that MoIE-CXR outperforms other models, including BB. Recall that MoIE-CXR refers to the mixture of all interpretable experts, excluding any residuals. As MoIE-CXR specializes in various subsets of data, it effectively discovers sample-specific classifying concepts and achieves superior performance. In general, MoIE-CXR exceeds the interpretable-by-design baselines (CEM, CBM, and CBM + ELEN) by a fair margin (on average, at least $\sim 10\% \uparrow$), especially for pneumonia and pneumothorax where the number of samples with the disease is significantly less ($\sim 750/24000$ in the testset). To compare the performance on the entire dataset, we additionally report MoIE-CXR+R, the mixture of interpretable experts with the final residual in Tab.2. MoIE-CXR+R outperforms the interpretable-by-design models and yields comparable performance as BB. The residualized PCBM baseline, *i.e.,* PCBM-h, performs similarly to MoIE-CXR+R. PCBM-h rectifies the interpretable PCBM's mistakes by learning the residual with the complete dataset to resemble BB's performance. However, the experts and the final residual approximate BB's interpretable and uninterpretable fractions,

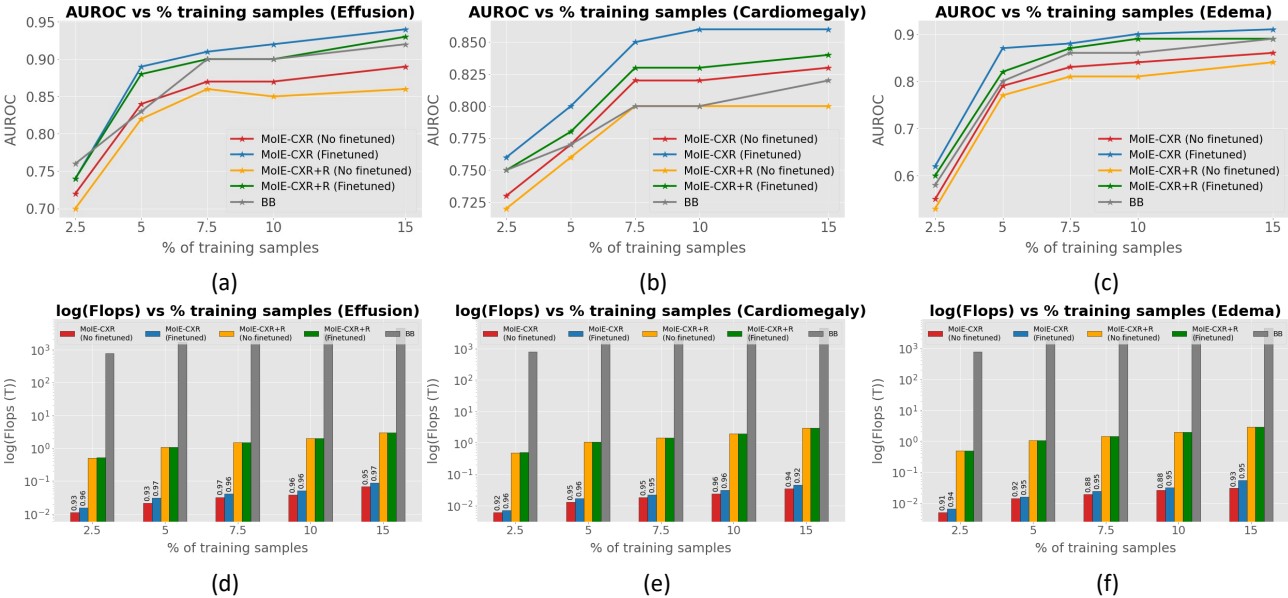

*Figure 4.* Transferring the first 3 experts of MoIE-CXR trained on MIMIC-CXR to Stanford-CXR. With varying % of training samples of Stanford CXR, **(a-c):** reports AUROC of the test sets, **(d-g)** reports computation costs in terms of $\log$ (Flops (T)). We report the coverages in Stanford-CXR on top of the "finetuned" and "No finetuned" variants of MoIE-CXR (red and blue bars) in **(d-g)**.

respectively. In each iteration, the residual focuses on the samples not covered by the respective expert to create BB for the next iteration and likewise. As a result, the final residual in MoIE-CXR+R covers the "hardest" examples, reducing its overall performance relative to MoIE-CXR. Also, Table 3 shows that MoIE outperforms the posthoc baselines' performance for the HAM10000 and SIIM-ISIC datasets. As mentioned earlier, interpretable by-design baselines are not possible for these datasets as we extract the concepts from Derm7pt using the embeddings from the BB.

### 4.1.4. APPLYING MOIE-CXR TO THE UNSEEN DOMAIN WITH LIMITED DATA

In this experiment, we utilize Algorithm 1 to transfer MoIE-CXR trained on MIMIC-CXR dataset to Stanford Chexpert (Irvin et al., 2019) dataset for the diseases – effusion, cardiomegaly and edema. Using 2.5%, 5%, 7.5%, 10%, and 15 % of training data from the Stanford Chexpert dataset, we employ two variants of MoIE-CXR where we (1) train only the selectors ($\pi$) without finetuning the experts ($g$) ("No finetuned" variant of MoIE-CXR in Fig. 4), and (2) finetune $\pi$ and $g$ jointly for only 5 epochs ("Finetuned" variant of MoIE-CXR and MoIE-CXR + R in Fig. 4). Finetuning $\pi$ is essential to route the target domain samples to the appropriate expert. As later experts cover the "harder" samples of MIMIC-CXR, we only transfer the experts of the first three iterations (refer to Fig. 3). To ensure a fair comparison,

we finetune (both the feature extractor $\Phi$ and classifier $h^0$) BB: $f^0 = h^0 \circ \Phi$ of MIMIC-CXR with the same training data of Stanford Chexpert for 5 epochs. Throughout this experiment, we fix $\Phi$ while finetuning the final residual in MoIE+R as stated in Eq. 4. Fig. 4 displays the performances of different models and the computation costs in terms of Flops. The Flops are calculated as, Flop of (forward propagation + backward propagation) $\times$ (total no. of batches) $\times$ (no of training epochs). The finetuned MoIE-CXR outperforms the finetuned BB (on average $\sim 5\% \uparrow$ for effusion and cardiomegaly). As experts are simple models (Barbiero et al., 2022) and accept only low dimensional concept vectors compared to BB, the computational cost to train MoIE-CXR is significantly lower than that of BB (Fig. 4 (d-f)). Specifically, BB requires $\sim 776$T flops to be finetuned on 2.5% of the training data of Stanford CheXpert, whereas MoIE-CXR requires $\sim 0.0065$T flops. As MoIE-CXR discovers the sample-specific domain-invariant concepts, it achieves such high performance with low computational cost than BB.

## 5. Conclusion

This paper proposes a novel interpretable method that identifies instance-specific concepts without losing the performance of the Blackbox and is effectively fine-tuned in an unseen target domain with no concept annotation, limited

labeled data, and minimal computation cost. The captured concepts may not showcase a causal effect that can be explored in the future.

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

# A. Appendix

## A.1. Code

Refer to the url https://github.com/batmanlab/MICCAI-2023-Route-interpret-repeat-CXRs for the code.

## A.2. Background of First-order logic (FOL) and Neuro-symbolic-AI

FOL is a logical function that accepts predicates (concept presence/absent) as input and returns a True/False output being a logical expression of the predicates. The logical expression, which is a set of AND, OR, Negative, and parenthesis, can be written in the so-called Disjunctive Normal Form (DNF) (Mendelson, 2009). DNF is a FOL logical formula composed of a disjunction (OR) of conjunctions (AND), known as the "sum of products".

Neuro-symbolic AI is an area of study that encompasses deep neural networks with symbolic approaches to computing and AI to complement the strengths and weaknesses of each, resulting in a robust AI capable of reasoning and cognitive modeling (Belle, 2020). Neuro-symbolic systems are hybrid models that leverage the robustness of connectionist methods and the soundness of symbolic reasoning to effectively integrate learning and reasoning (Garcez et al., 2015; Besold et al., 2017).

## A.3. Learning the concepts

As discussed in Section 2, $f^0 : \mathcal{X} \to \mathcal{Y}$ is a pre-trained Blackbox. Also, $f^0(.) = h^0 \circ \Phi(.)$. Here, $\Phi : \mathcal{X} \to R^l$ is the image embeddings, transforming the input images to an intermediate representation and $h^0 : R^l \to \mathcal{Y}$ is the classifier, classifying the output $\mathcal{Y}$ using the embeddings, $\Phi$. Our approach is applicable for both datasets with and without human-interpretable concept annotations. For datasets with the concept annotation $\mathcal{C} \in \mathbb{R}^{N_c}$ ($N_c$ being the number of concepts per image $\mathcal{X}$), we learn $t : R^l \to \mathcal{C}$ to classify the concepts using the embeddings. Per this definition, $t$ outputs a scalar value $c$ representing a single concept for each input image. We adopt the concept learning strategy in PosthocCBM (PCBM) (Yuksekgonul et al., 2022) for datasets without concept annotation. Specifically, we leverage a set of image embeddings with the concept being present and absent. Next, we learn a linear SVM ($t$) to construct the concept activation matrix (Kim et al., 2017) as $Q \in \mathbb{R}^{N_c \times l}$. Finally we estimate the concept value as $c = \frac{<\Phi(x), q^i>}{||q_i||_2^2} \in \mathbb{R}$ utilizing each row $q^i$ of $Q$. Thus, the complete tuple of $j^{th}$ sample is $\{x_j, y_j, c_j\}$, denoting the image, label, and learned concept vector, respectively.

## A.4. Optimization

In this section, we will discuss the loss function used in distilling the knowledge from the blackbox to the symbolic model. We remove the superscript $k$ for brevity. We adopted the optimization proposed in (Geifman & El-Yaniv, 2019).Specifically, we convert the constrained optimization problem in Equation (5) as

$$\mathcal{L}_s = \mathcal{R}(\pi, g) + \lambda_s \Psi(\tau - \zeta(\pi)) \tag{6}$$
$$\Psi(a) = \max(0, a)^2,$$

where $\tau$ is the target coverage and $\lambda_s$ is a hyperparameter (Lagrange multiplier). We define $\mathcal{R}(.)$ and $\mathcal{L}_{g,\pi}(.)$ in Equation (1) and Equation (3) respectively. $\ell$ in Equation (3) is defined as follows:

$$\ell(f, g) = \ell_{distill}(f, g) + \lambda_{lens} \sum_{i=1}^{r} \mathcal{H}(\beta^i), \tag{7}$$

where $\lambda_{lens}$ and $\mathcal{H}(\beta^i)$ are the hyperparameters and entropy regularize, introduced in (Barbiero et al., 2022) with $r$ being the total number of class labels. Specifically, $\beta^i$ is the categorical distribution of the weights corresponding to each concept. To select only a few relevant concepts for each target class, higher values of $\lambda_{lens}$ will lead to a sparser configuration of $\beta$. $\ell$ is the knowledge distillation loss (Hinton et al., 2015), defined as

$$\ell(f, g) = (\alpha_{KD} * T_{KD} * T_{KD}) KL\big(\text{LogSoftmax}(g(.)/T_{KD}), \text{Softmax}(f(.)/T_{KD})\big) + \tag{8}$$
$$(1 - \alpha_{KD}) CE\big(g(.), y\big),$$

where $T_{KD}$ is the temperature, CE is the Cross-Entropy loss, and $\alpha_{KD}$ is relative weighting controlling the supervision from the blackbox $f$ and the class label $y$.

As discussed in (Geifman & El-Yaniv, 2019), we also define an auxiliary interpretable model using the same prediction task assigned to $g$ using the following loss function

$$\mathcal{L}_{aux} = \frac{1}{m} \sum_{j=1}^{m} \ell_{distill}(f(\boldsymbol{x_j}), g(\boldsymbol{c_j})) + \lambda_{lens} \sum_{i=1}^{r} \mathcal{H}(\beta^i), \tag{9}$$

which is agnostic of any coverage. $\mathcal{L}_{aux}$ is necessary for optimization as the symbolic model will focus on the target coverage $\tau$ before learning any relevant features, overfitting to the wrong subset of the training set. The final loss function to optimize by g in each iteration is as follows:

$$\mathcal{L} = \alpha \mathcal{L}_f + (1 - \alpha)\mathcal{L}_{aux}, \tag{10}$$

where $\alpha$ is the can be tuned as a hyperparameter. Following (Geifman & El-Yaniv, 2019), we also use $\alpha = 0.5$ in all of our experiments.

### A.5. Algorithm

Algorithm 2 explains the overall training procedure of our method. Figure 5 displays the architecture of our model in iteration $k$.

### A.6. Dataset

**HAM10000** HAM10000 ((Tschandl et al., 2018)) is a classification dataset aiming to classify a skin lesion benign or malignant. Following (Daneshjou et al., 2021), we use Inception (Szegedy et al., 2015) model, trained on this dataset as the blackbox $f^0$. We follow the strategy in (Lucieri et al., 2020) to extract the 8 concepts from the Derm7pt ((Kawahara et al., 2018)) dataset.

**SIIM-ISIC** To test a real-world transfer learning use case, we evaluate the model trained on HAM10000 on a subset of the SIIM-ISIC(Rotemberg et al., 2021)) Melanoma Classification dataset. We use the same concepts described in the HAM10000 dataset.

**MIMIC-CXR** We use 220,763 frontal images from the MIMIC-CXR dataset (Johnson et al.) aiming to classify effusion. We obtain the anatomical and observation concepts from the RadGraph annotations in RadGraph's inference dataset ((Jain et al., 2021)), automatically generated by DYGIE++ ((Wadden et al., 2019)). We use the test-train-validation splits from (Yu et al., 2022) and Densenet121 (Huang et al., 2017) as the blackbox $f^0$.

### A.7. Architectural details of symbolic experts and hyperparameters

Table 4 demonstrates different settings to train the Blackbox of MIMIC-CXR. For the Blackbox of HAM10000, we follow the steps in (Yuksekgonul et al., 2022). To train $t$, for MIMIC-CXR and HAM10000, we flatten out the feature maps from the last convolutional block. Table 5, Table 6 enumerate all the different settings to train the interpretable experts for HAM10000, and MIMIC-CXR respectively. All the residuals in different iterations follow the same settings as their Blackbox counterparts.

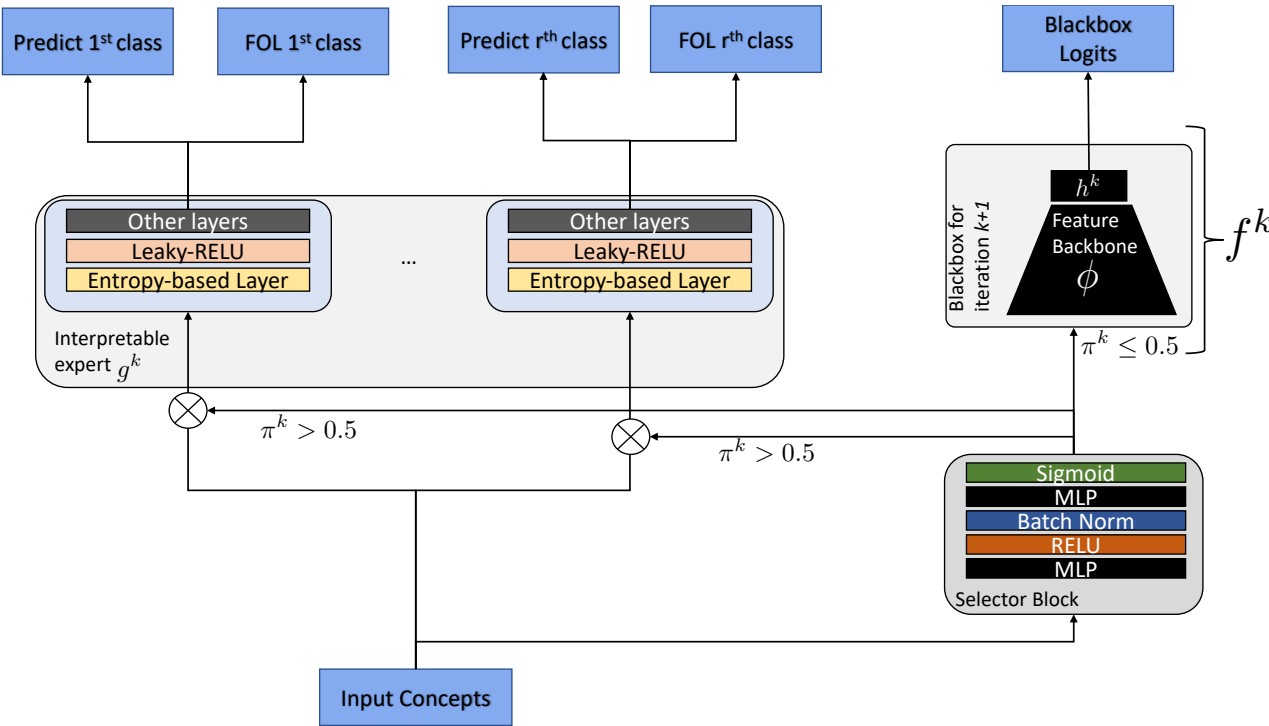

*Figure 5.* Architecture of MoIE. In an iteration $k$ during inference, the selector routes the samples to go through the interpretable expert $g^k$ if the probability $\pi^k \geq 0.5$. If $\pi^k < 0.5$, the selector routes the samples, through $f^k$, the Blackbox for iteration $k+1$. Note $f^k = h^k(\Phi(.)$ is an approximation of the residual $r^k = f^{k-1} - g^k$.

## A.8. More results

## A.9. Qualitative results of FOLs for MIMIC-CXR

Figure 6 demonstrates that MoIE-CXR captures diverse explanations for pneumonia and edema. Also Figure 7 reports the FOL explanations for MoIE for HAM10000 and SIIM-ISIC datasets respectively.

## A.10. Later experts cover harder samples

Figure 8 and Figure 9 demonstrate further examples that later experts of MoIE and MoIE-CXR cover harder samples for HAM10000 and MIMIC-CXR datasets respectively.

---

**Algorithm 2** *Route, interpret* and *repeat* algorithm to generate FOL explanations locally.

---

1: **Input:** Complete tuple: $\{x_j, y_j, c_j\}_{j=1}^n$; initial blackbox $f^0 = h^0(\Phi(.))$; K as the total iterations; Coverages $\tau_1, \ldots, \tau_K$.
2: **Output:** Sparse mixture of experts and their selectors $\{g^k, \pi^k\}_{k=1}^K$ and the final residual $f^K = h^K(\Phi(.))$
3: Fix $\Phi$.
4: **for** $k = 1 \ldots K$ **do**
5:     Fix $\pi^1 \ldots \pi^{k-1}$.
6:     Minimize $\mathcal{L}^k$ using equation 10 to learn $\pi^k$ and $g^k$.
7:     Calculate $r^k = f^{k-1}(.) - g^k(.)$
8:     Minimize equation 4 to learn $f^k(.)$, the new blackbox for the next iteration $k+1$.
9: **end for**
10: **for** $k = 1 \ldots K$ **do**
11:     **for** sample $j$ in `test-set` **do**
12:         **repeat**
13:             Initialize `sub_select_concept` $= True$
14:             Initialize the `percentile_threshold` $= 99$.
15:             Retrieve the predicted class label of sample $j$ from the expert $k$ as: $\hat{y}_j = g^k(c_j)$
16:             Create a mask vector $m_j$. $m_j[i] = 1$ if $\tilde{\alpha}[\hat{y}_j][i] \geq$ percentile($\tilde{\alpha}[\hat{y}_j]$, `percentile_threshold`) and 0 otherwise. Specifically, the $i^{th}$ entry in $m_j$ is one if the $i^{th}$ value of the attention score $\tilde{\alpha}[\hat{y}_j]$ is greater than (`percentile_attention`)$^{th}$ percentile.
17:             Subselect the concept vector as $\tilde{c}_j$ as: $\tilde{c}_j = c_j \odot m_j$
18:             **if** $g^k(\tilde{c}_j) \neq \hat{y}_j$ **then**
19:                 `percentile_threshold` $=$ `percentile_threshold` $- 1$
20:                 `sub_select_concept` $= false$
21:             **end if**
22:         **until** `sub_select_concept` is $True$
23:         Using the subselected concept vector $\tilde{c}_j$, construct the FOL expression of the $j^{th}$ sample as suggested by (Barbiero et al., 2022).
24:     **end for**
25: **end for**

---

*Table 4.* Hyperparameter setting of different convolution-based Blackboxes used by the diseases in MIMIC-CXR considering each to be a separate binary classification problem

| Setting | MIMIC-CXR |
|---|---|
| Backbone | DenseNet-121 |
| Pretrained on ImageNet | True |
| Image size | 512 |
| Learning rate | 0.01 |
| Optimization | SGD |
| Weight-decay | 0.0001 |
| Epcohs | 50 |
| Layers used as $\Phi$ | till $4^{th}$ DenseNet Block |
| Flattening type for the input to $t$ | Flatten |

*Table 5.* Hyperparameter setting of the different interpretable experts ($g$) for the dataset HAM10000

| Settings based on dataset | Expert1 | Expert2 | Expert3 | Expert4 | Expert5 | Expert6 |
|---|---|---|---|---|---|---|
| HAM10000 (Inception-V3) | | | | | | |
| + Batch size | 32 | 32 | 32 | 32 | 32 | 32 |
| + Coverage ($\tau$) | 0.4 | 0.2 | 0.2 | 0.2 | 0.1 | 0.1 |
| + Learning rate | 0.01 | 0.01 | 0.01 | 0.01 | 0.01 | 0.01 |
| + $\lambda_{lens}$ | 0.0001 | 0.0001 | 0.0001 | 0.0001 | 0.0001 | 0.0001 |
| + $\alpha_{KD}$ | 0.9 | 0.9 | 0.9 | 0.9 | 0.9 | 0.9 |
| + $T_{KD}$ | 10 | 10 | 10 | 10 | 10 | 10 |
| +hidden neurons | 10 | 10 | 10 | 10 | 10 | 10 |
| +$\lambda_s$ | 64 | 64 | 64 | 64 | 64 | 64 |
| + $T_{lens}$ | 0.7 | 0.7 | 0.7 | 0.7 | 0.7 | 0.7 |

*Table 6.* Hyperparameters of interpretable experts ($g$) for the dataset MIMIC-CXR.

| Hyperparameter | Effusion | Cardiomegaly | Pneumothorax | Pneumonia | Edema |
|---|---|---|---|---|---|
| Batch size | 1028 | 1028 | 1028 | 1028 | 1028 |
| Learning rate | 0.01 | 0.01 | 0.01 | 0.01 | 0.01 |
| $\lambda_{lens}$ | 0.0001 | 0.0001 | 0.0001 | 0.0001 | 0.0001 |
| $\alpha_{KD}$ | 0.99 | 0.99 | 0.99 | 0.99 | 0.99 |
| $T_{KD}$ | 20 | 20 | 20 | 20 | 20 |
| hidden neurons | 30, 30 | 20, 20 | 20, 20 | 20, 20 | 20, 20 |
| $\lambda_s$ | 96 | 1024 | 256 | 256 | 128 |
| E-Lens ($T_{lens}$) | 7.6 | 7.6 | 10 | 10 | 7.6 |
| # Expers ($T_{lens}$) | 5 | 4 | 5 | 4 | 5 |

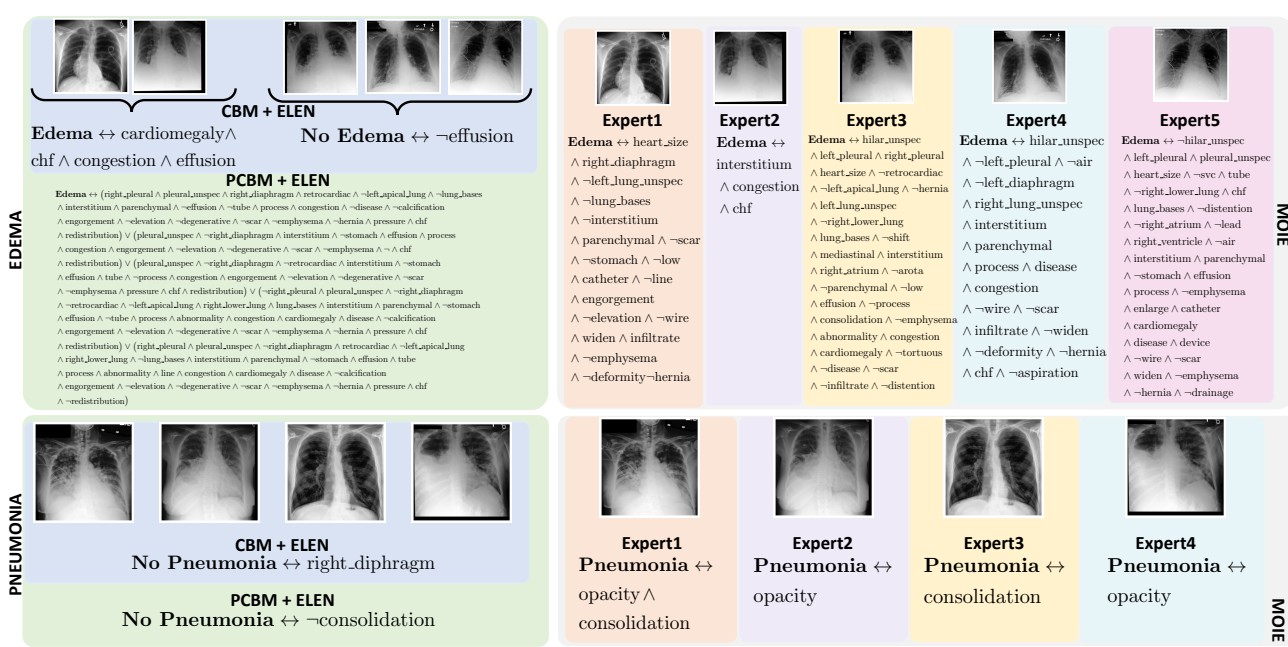

*Figure 6.* Qualitative comparison of MoIE discovered concepts with the baseline for edema and pneumonia.

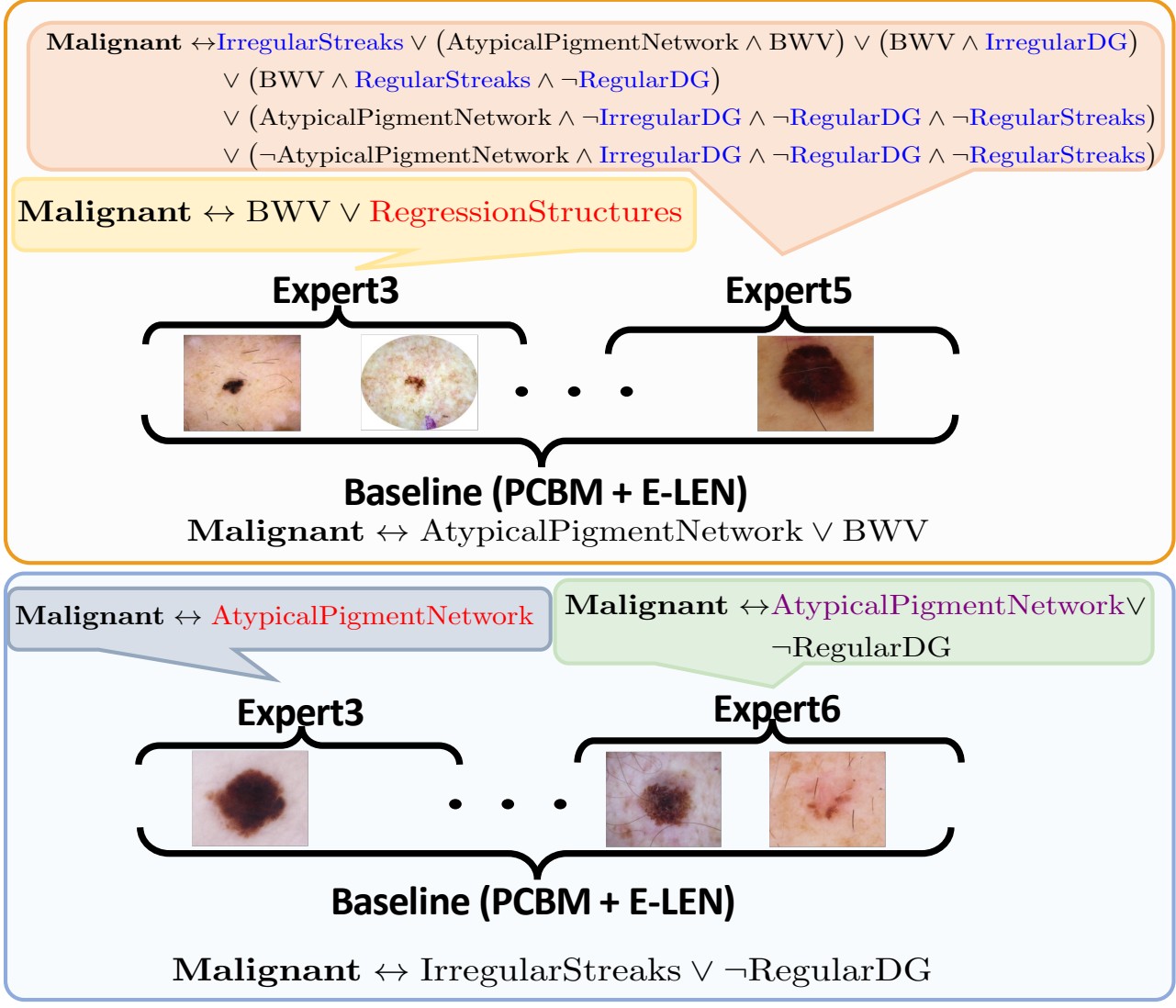

*Figure 7.* MoIE identifies diverse concepts for specific subsets of a class, unlike the generic ones by the baselines. We compare FOL explanations by MoIE with the PCBM + E-LEN baselines for HAM10000 (**top**) and ISIC (**down**) to classify Malignant lesions. We highlight unique concepts for experts 3, 5, and 6 in *red*, *blue*, and *violet*, respectively. For brevity, we combine the local FOLs for each expert for the samples covered by them.

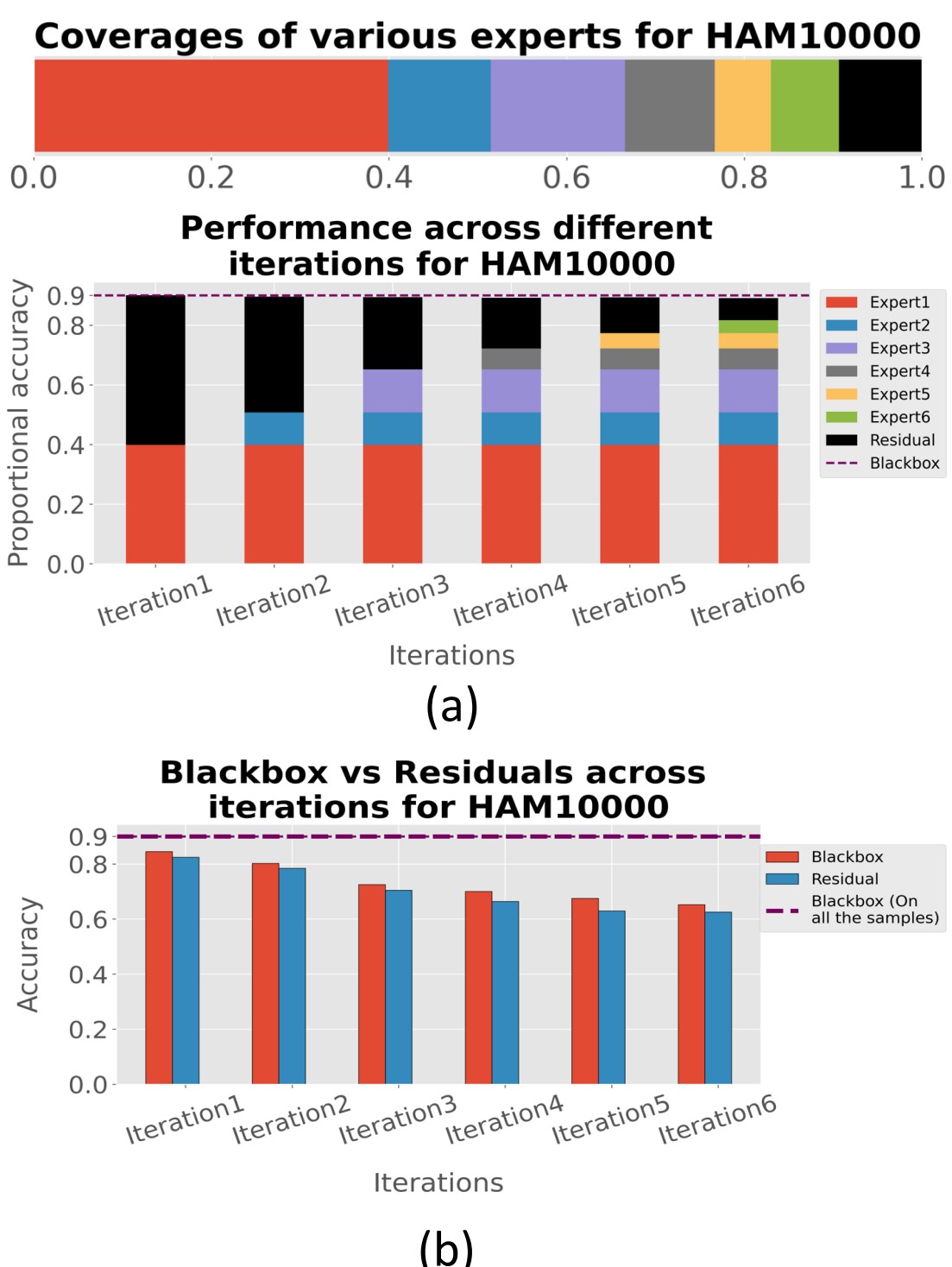

*Figure 8.* Performance of experts and residuals across iterations. **(a-c):** Coverage and proportional accuracy of the experts and residuals of MoIE for HAM10000 dataset. **(d-f):** Routing the samples covered by MoIE to the initial $f^0$, we compare the performance of the residuals with $f^0$.

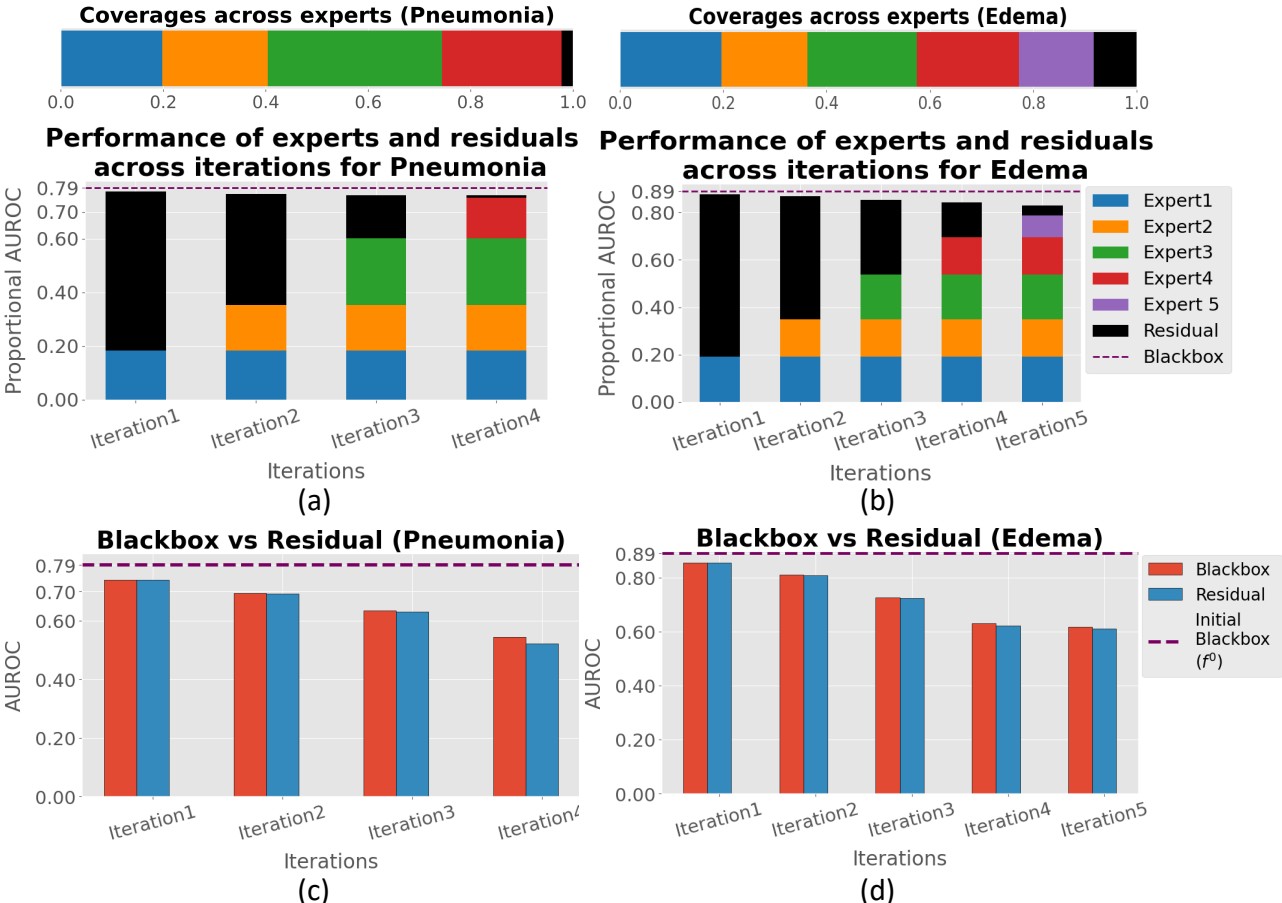

*Figure 9.* **(a-c):** Coverage and proportional AUROC of the experts and residuals of MoIE-CXR for the diseases pneumonia and edema of MIMIC-CXR dataset. **(d-f):** Routing the samples covered by MoIE-CXR to the initial $f^0$, we compare the performance of the residuals with $f^0$.

