# OpenReview forum: "Bridging the Gap: From Post Hoc Explanations to Inherently Interpretable Models for Medical Imaging"
_ICML.cc/2023/Workshop/IMLH — IMLH 2023 Poster_

### Official Review · Reviewer_o6tf · 2023-06-10
**Novel Idea, could use more clarity**

**Rating:** 7
**Confidence:** 3

**Review:**

I am a little confused as to how the symbolic models work. If you have more space, could you explain them in more detail?
In figure 2, MOIE describes subsets of logic classes. Can the authors elucidate if these sub classes make sense or if they reveal some kind of pattern?
Some sub classes also seem to overlap. Like effusion experts 3 and 4. Is this redundancy common throughout the subjects? Are some of the experts actually not necessary?

---

### Official Review · Reviewer_yfL2 · 2023-06-15
**This paper proposes a novel strategy to train separate models (experts) to understand and explain different parts of a given dataset. This is done by starting with a black box model, and then iteratively split the data into experts and a residual network.  The authors show increased performance for several datasets, when comparing to other state of the art procedures such as concept bottleneck models.**

**Rating:** 7
**Confidence:** 4

**Review:**

The current paper is clearly written (except for a few chosen words, please see below), with detailed mathematical notation and derivations, resulting in a high quality paper. The work is original, and the work may significantly contribute to better concept mapping between data and models.

Pros:

•	Code and data is publicly available

•	Clear mathematical notation

Cons:

•	Please consider to use other words than ‘blur’ and ‘carve out’ and replace them with a more scientific description

•	The authors could potentially provide a little bit more of a justification for defining the criteria for defining the setup of iterations. For example, what does it mean that an expert covers enough samples, otherwise the process is stopped? The same goes for the definition of the final residual underperforming at a certain threshold?

---

### Official Review · Reviewer_bS1V · 2023-06-16
**blur the distinction between post hoc explanation and constructing interpretable model**

**Rating:** 7
**Confidence:** 4

**Review:**

Summary and Strength:
The paper proposes a method to blur the distinction between post hoc explanation of a Blackbox (BB) model and constructing interpretable models. The proposed method begins with a BB model and iteratively carves out a mixture of interpretable experts and a residual network. Each interpretable model specializes in a subset of samples and explains them using First Order Logic (FOL), while the remaining samples are routed through a flexible residual network. The method is repeated until all the interpretable models explain the desired proportion of data. The results of extensive experiments show that the proposed approach can identify a diverse set of instance-specific concepts without compromising the performance of the BB model, identify the relatively "harder" samples to explain via residuals, and be transferred to an unknown target domain with limited data efficiently.

Weakness:
The paper may have a weakness in that all experiments are conducted using 2D data, which may limit the generalizability of the proposed method to 3D datasets. It is recommended that the authors validate the effectiveness of their method on 3D datasets to ensure its broader applicability.

---

### Meta-Review · Area_Chair_E4kY · 2023-06-20

**Recommendation:** Accept (Poster)
**Confidence:** 5

**Metareview:**

All three reviewers expressed positive opinions about this paper, appreciating its clarity and extensive experimentation. I kindly ask the authors to carefully consider the identified shortcomings and ensure that these issues are addressed in the final version.

---

### Decision · Program_Chairs · 2023-06-20

Accept (Poster)